# CROSS-MODAL GRAPH CONTRASTIVE LEARNING WITH CELLULAR IMAGES

## ABSTRACT

Constructing discriminative representations of molecules lies at the core of a number of domains such as drug discovery, material science, and chemistry. State-of-the-art methods employ graph neural networks (GNNs) and self-supervised learning (SSL) to learn the structural representations from unlabeled data, which can then be fine-tuned for downstream tasks. Albeit powerful, these methods that are pre-trained solely on molecular structures cannot generalize well to the tasks involved in intricate biological processes. To cope with this challenge, we propose using high-content cell microscopy images to assist in learning molecular representation. The fundamental rationale of our method is to leverage the correspondence between molecular topological structures and the caused perturbations at the phenotypic level. By including cross-modal pre-training with different types of contrastive loss functions in a unified framework, our model can efficiently learn generic and informative representations from cellular images, which are complementary to molecular structures. Empirical experiments demonstrated that the model transfers non-trivially to a variety of downstream tasks and is often competitive with the existing SSL baselines, e.g., a 15.4% absolute Hit@10 gains in graph-image retrieval task and a 4.0% absolute AUC improvements in clinical outcome predictions. Further zero-shot case studies show the potential of the approach to be applied to real-world drug discovery.

## 1 INTRODUCTION

Learning discriminative representations of molecules is a fundamental task for numerous applications such as molecular property prediction, de novo drug design and material discovery Wu et al. (2018). Since molecular structures are essentially topological graphs with atoms and covalent bonds, graph representation learning can be naturally introduced to capture the representation of molecules Duvenaud et al. (2015); Xu et al. (2019); Song et al. (2020); Ying et al. (2021). Despite the fruitful progress, graph neural networks (GNNs) are known to be data-hungry, i.e., requiring a large amount of labeled data for training. However, task-specific labeled data are far from sufficient, as wet-lab experiments are resource-intensive and time-consuming. As a result, training datasets in chemistry and drug discovery are typically limited in size, and neural networks tend to overfit them, leading to poor generalization capability of the learned representations.

Inspired by the fruitful achievements of the self-supervision learning in computer vision Chen et al. (2020a); He et al. (2020) and natural language processing Devlin et al. (2018); Radford et al. (2018), there have been some attempts to extend self-supervised schemes to molecular representation learning and lead to remarkable results Hu et al. (2020b); Xu et al. (2021); You et al. (2020; 2021); Rong et al. (2020); Velickovic et al. (2019). The core of self-supervised learning lies in establishing meaningful pre-training objectives to harness the power of large unlabeled data. The pre-trained neural networks can then be used to fine-tune for small-scale downstream tasks.

However, pre-training on molecular graph structures remains a stiff challenge. One of the limitations of current approaches is the lack of domain knowledge in chemistry or chemical synthesis. Recent studies have pointed out that pre-trained GNNs with random node/edge masking gives limited improvements and often lead to negative transfer on downstream tasks Hu et al. (2020b); Stärk et al. (2021), as the perturbations actions on graph structures can hurt the structural inductive bias of molecules. Furthermore, molecular modeling tasks often require predicting the binding/interaction

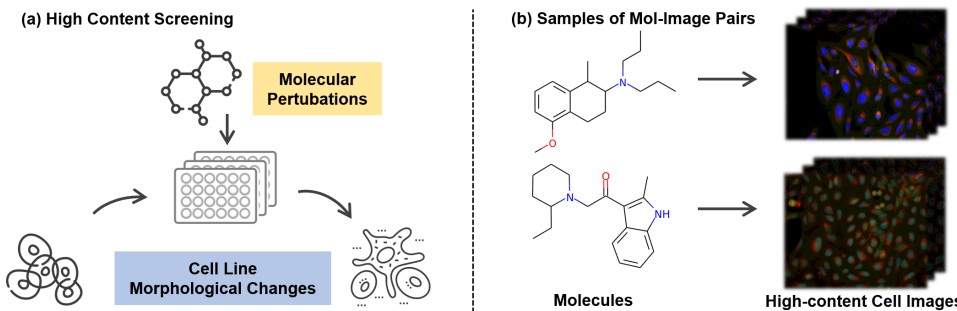

Figure 1: (a) Illustration of high-content screening experimental protocol. (b) Samples of molecule-images pairs. Note that one molecule produces multiple images.

between molecules and other biological entities (e.g., RNA, proteins, pathways), and further generalizing to the phenotypic/clinical outcomes caused by these specific bindings Kuhn et al. (2016). Self-supervised learning methods that solely manipulate molecular structures struggle to handle downstream tasks that involve complex biological processes, limiting their practicality in a wide range of drug discovery applications.

To this end, we propose using high-content cell microscopy images to assist in learning molecular representation, extending the molecular representation beyond chemical structures and thus, improving the generalization capability. High-content cell microscopy imaging (HCI) is an increasingly important biotechnology in recent years in the domain of drug discovery and system biology Chandrasekaran et al. (2021); Bray et al. (2016). As shown in Figure 1, small molecules enter into cells and affect their biological functions and pathways, leading to morphological changes in cell shape, number, structure, etc., that are visible in microscopy images after staining. Analysis and modeling based on these high-content images have shown great success in molecular bioactivity prediction Hofmarcher et al. (2019), mechanism identification Caicedo et al. (2022), polypharmacology prediction Chow et al. (2022), etc. Thus, we hypothesize that this phenotypic modality has complementary strengths with molecular structures to make enhanced representations and thus benefit the downstream tasks involved in intricate biological processes.

However, building connections between molecular structures and high-content cellular images is a challenging task that highlights representative, unsolved problems in cross-modal learning. The first challenge comes from the unclear fine-grained correspondence between molecules and images. Unlike conventional cross-modal paired data such as caption and picture, the patterns of the molecule are not directly reflected in the cellular image, thus preventing us from using traditional cross-modal encoders for alignment. The second challenge arises from the noise and batch effect of the cellular data. For example, cellular images obtained from the same molecule can vary considerably. Existing cross-modal pre-training objectives may overfit the noisy images and decrease the model's generalization ability.

Herein, we propose Molecular graph and hIgh content imaGe Alignment (MIGA), a novel cross-modal graph-and-image pre-training framework to address the above issues. We first encode the molecule and cell microscopy images independently with a molecular graph encoder and an image encoder. Then we align the graph embeddings with the image embeddings through three contrastive modules: graph-image contrastive (GIC) learning, masked graph modeling (MGM) and generative graph-image matching (GGIM). Specifically, (i) GIC encourages high similarity between the latent embeddings of matched graph-image pairs while pushing those of non-matched pairs apart; (ii) MGM, a local cross-modal module that utilizes both the observed (unmasked) graph and the cell image to predict the masked molecular patterns and (iii) GGIM aims to further distinguish the hard negative graph-image pairs that share similar global semantics but differ in fine-grained details. The three modules are complementary and thus the combination of these modules can 1) make it easier for the encoders to perform cross-modal learning by capturing structural and localized information; (2) learn a common low-dimensional space to embed graphs and images that are biologically meaningful. Enabled by the massive publicly available high-content screening data Bray et al. (2017), we establish a novel cross-modal benchmark dataset that contains 750k molecular graph-cellular image pairs. To evaluate models on this benchmark, we propose a new biological meaningful retrieval task specific to

graph-image cross-modal learning. We also include existing clinical outcome prediction and property prediction tasks to further assess the learned representations. Extensive experiments demonstrate that the cross-modal representations learned by our proposed model, MIGA, can benefit a wide range of downstream tasks that require extensive biological priors. For example, MIGA achieves a 15.4% absolute Hit@10 gain in graph-image retrieval task and a 4.0% absolute AUC improvement in clinical outcome predictions, respectively, over existing state-of-the-art methods.

## 2 RELATED WORK

**Self-supervised learning on graphs.** Graph self-supervised pre-learning attempts to obtain supervision in unlabelled data to learn meaningful representations that can be further transferred to downstream tasks You et al. (2020); Hu et al. (2020b;c); Xu et al. (2021); You et al. (2021); Rong et al. (2020); Velickovic et al. (2019); Sun et al. (2019); Zhang et al. (2020); Liu et al. (2021a); Stärk et al. (2021). In general, these methods fall into two categories: contrastive-based methods and generative-based methods Liu et al. (2021b). The former aims to generate different views from the original graph and perform intra-graph contrastive learning Hu et al. (2020b); Xu et al. (2021); You et al. (2020; 2021); Velickovic et al. (2019), while the latter ones are trained in a supervised manner to generate masked sub-patterns or attributes at the inter-graph level Hu et al. (2020b); Rong et al. (2020); Zhang et al. (2020). These approaches achieve remarkable performance on molecular graph representation tasks, but lack the ability to predict the complex properties involved in intricate biological processes.

**Cross-modal pre-training.** Pre-training strategies for multi-modal tasks have attracted massive attention, with most of these efforts targeting Visual-Language representation learning. Most of them can be grouped into two categories. The first category is to use multi-modal encoders to capture the interaction between image and text embeddings Su et al. (2019); Lu et al. (2020); Chen et al. (2020b); Li et al. (2020); Huang et al. (2021); Zhang et al. (2021). Approaches in this category achieve remarkable performance, but most of them require high-quality images and pre-trained object detectors. The other category focuses on learning independent decoders for different modalities Radford et al. (2021); Jia et al. (2021); Xu et al. (2022); Taleb et al. (2022). For instance, CLIP Radford et al. (2021) learns pairwise relationships between language and images by performing pre-training on a large amount of web data using independent encoders connected by contrastive losses. More recently, CLOOME Sanchez-Fernandez et al. (2022) employs a CLIP-like loss to pre-train the cellular image encoder with molecular fingerprints. Different from their focus on the image encoder, MIGA aims at extracting information from cellular images to make enhanced molecular representations and designed two complementary cross-modal contrastive losses for information interaction in an end-to-end manner.

## 3 METHODS

At the core of our method is the idea of infusing structural representations with biological perception by building the connections between molecular graph and their induced morphological features.

To achieve this, as illustrated in Figure 2, given pairs of graph and image, we employ two independent encoders (a GNN $f^g$ and a CNN $f^i$) to produce the representations of a molecular graph $G$ and a cellular image $I$, and align them with inter- and intra- contrastive losses. This cross-modal framework pulls the matched graph-image pairs together and contrasts the unmatched pairs apart. After pre-training, we use the output representations to make zero-shot predictions or fine-tune the networks on downstream tasks. This cross-modal learning process can also be interpreted as morphological information being passed through the convolutional neural network to the graph neural network in a knowledge distillation manner Tian et al. (2019).

In the following sections, we first introduce the structural and image encoders for cross-modal learning (section 3.1). Then we depict the details of contrastive training objectives (section 3.2), followed by the experimental setting (section 4.1) and results (section 4.2).

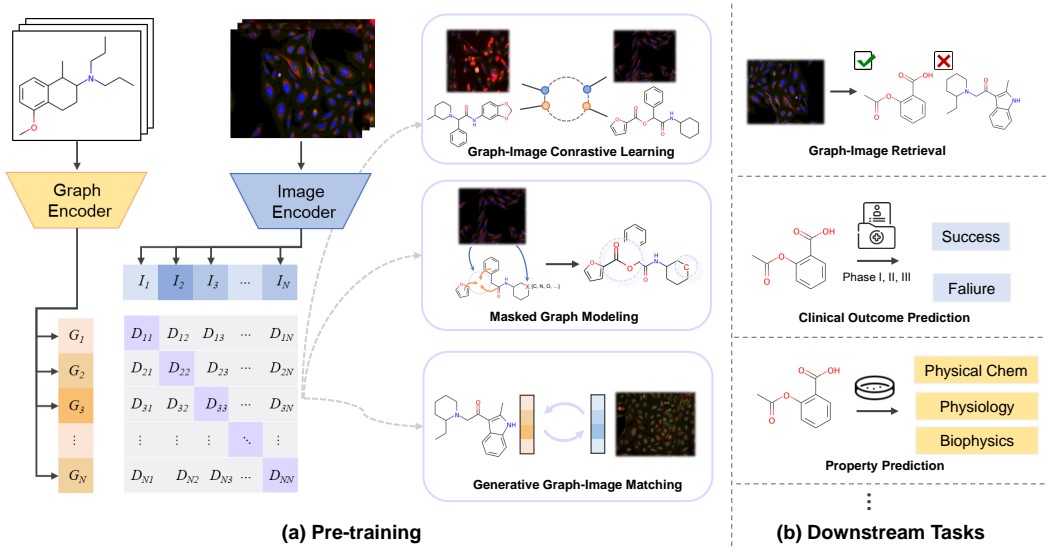

**(a) Pre-training**   **(b) Downstream Tasks**

Figure 2: Overview of our method, MIGA. Molecular graph and cellular image representations are jointly learned from pairwise data. We used three complementary contrastive objectives to perform cross-modal pre-training. The learned representations can be used for graph-image or graph-only downstream task transfer.

## 3.1 STRUCTURE AND IMAGE ENCODERS

**Structural encoder.** A compound structure can be represented as an attributed graph $G = (\mathcal{V}, \mathcal{E})$, where $|\mathcal{V}| = n$ denotes a set of $n$ atoms (nodes) and $|\mathcal{E}| = m$ denotes a set of $m$ bonds (edges). We represent $X_v \in \mathbb{R}^{n \times d_n}$ for the node attributes with $d_n$ as the the feature dimension of node and $E_{uv} \in \mathbb{R}^{m \times d_e}$ for the edge attributes with $d_e$ as the feature dimension of edge. A graph neural network (GNN) $f^g$ learns to embed an attributed graph $G$ into a feature vector $z_G$. We adopt the Graph Isomorphism Network (GIN) from Xu et al. (2019), where the node and edge attributes are propagated at each iteration. Formally, the k-th iteration of a GNN is:

$$h_v^{(k)} = g_U^{(k)}(h_v^{(k-1)}, g_{AGG}^{(k)}\{(h_v^{(k-1)}, h_u^{(k-1)}, X_{uv}) : u \in \mathcal{N}(v)\}) \tag{1}$$

where $h_v^{(k)}$ are the representation of node $v$ at the $k$-th layer, $\mathcal{N}(v)$ is the neighbourhood set of node $v$, $h_v^{(0)}$ is initialised with $X_v$ encoding its atom properties. $g_{AGG}^{(k)}$ stands for the aggregation function and $g_U^{(k)}$ stands for the update function. After $K$ graph convolutions, $h^K$ have captured their $K$-hop neighbourhood information. Finally, a readout function is used to aggregate all node representations output by the $K$-th GNN layer to obtain the entire molecule's representation $z_G$:

$$z_G = \sum_{v \in \mathcal{V}} READOUT(h_v^{(K)}) \tag{2}$$

**Image encoder.** For the cellular image $I$, we first use residual convolutional neural networks (ResNet-34) He et al. (2016) as the basic feature extractor $f^i$ because it is lightweight, widely-adopted and have proven performance. We also implement DenseNet Huang et al. (2017), EfficientNet Tan & Le (2019) and the recently proposed Vision Transformer (ViT) Dosovitskiy et al. (2020) as comparisons. We strictly follow their original implementation with a small modification, namely adding an additional projection layer before the final output. These models are initialized with weights pre-trained on ImageNet Deng et al. (2009). Each input image $I$ is encoded into a one-dimensional feature vector $z_I$ for further fusion.

## 3.2 PRE-TRAINING FRAMEWORK

We pre-trained MIGA through three contrastive objectives: graph-image contrastive (GIC) learning, masked graph modeling (MGM) and generative graph-image matching (GGIM).

**Graph-image contrastive (GIC) learning** aims to pull embeddings of the matched molecule-image pairs together while pushing those of unmatched pairs apart by maximizing a lower bound on the mutual information (MI) between the molecular graph and cellular image for the positive pairs. We achieve this by minimizing a symmetric InfoNCE loss Van den Oord et al. (2018) to maximize a lower bound on $MI(G; I)$. Formally, the graph-image contrastive loss is defined as:

$$\mathcal{L}_{GIC} = -\frac{1}{2}\mathbb{E}_{p(G,I)}[\,log\frac{exp(s\langle z_G, z_I\rangle/\tau)}{\sum_{k\neq i}^{K} exp(s\langle z_G, z_I^k\rangle)} + log\frac{exp(s\langle z_I, z_G\rangle/\tau)}{\sum_{k\neq i}^{K} exp(s\langle z_I, z_G^k\rangle)}] \tag{3}$$

Here, the similarity function $s\langle z_G, z_I\rangle = \|f_v(z_G) - f_w(z_I)\|_2$, where $f_v$ and $f_w$ are two linear projection layers that embed representations to a common space. $\tau$ is a temperature parameter, $K$ is a set of negative image samples that not matched to $G$.

**Masked graph modeling (MGM).** To simultaneously leverage the intra-molecular graph information and strengthen the interaction between molecule and image, we further utilize both the image and the partial molecular graph to predict the masked sub-patterns. Following the masking strategies of Hu et al. (2020b), we randomly mask the atom/bond attributes and constructed the context graphs for each molecular graph. We then use the surrounding partial graph structures along with the corresponding image information to predict the masked attributed subgraphs and the corresponding attributes. Our goal is to pre-train molecular GNN $f^g$ that can not only learn the context information of atoms in similar structures but also capture domain knowledge by learning the regularities of the node/edge attributes distributed over graph structure. Herein, we defined the masked molecular graph as $G^{msk}$ and the observed (unmasked) graph as $G^{obs}$. Therefore, the training graph-image pair $(G, I)$ has been transformed to $(G^{msk}, G^{obs}, I)$ and the objective of MGM is:

$$\mathcal{L}_{MGM} = \mathbb{E}_{(G^{msk}, G^{obs}, I)\sim D}[\log P(G^{msk}|(G^{obs}, I))] \tag{4}$$

where $(G^{msk}, G^{obs}, I)$ is a randomly sampled graph-image pair from the training set $D$. Thus, the MGM training is equivalent to optimizing the cross-entropy loss: :

$$\mathcal{L}_{MGM} = -\frac{1}{|D|}\sum_{(g_k^{msk}, g_k^{obs}, I_k)}^{D}\log[p(\hat{g}_k^{msk}|f_m(g_k^{obs}, z_{I_k}))] \tag{5}$$

where the $\hat{g}^{msk}$ is the prediction from the observed graph $g^{obs}$ and image embedding $z_I$. The function $f_m$ is the molecule attributes and context graphs prediction model. Our MLM training is minimized by a cross-entropy loss because $g^{msk}$ is a one-hot vocabulary distribution where the ground-truth attributes/subgraphs has a probability of 1.

**Generative Graph-image matching (GGIM)** aims to further distinguish the *hard* negative graph-image pairs that share similar global semantics but differ in fine-grained details. This is achieved by the combination of a graph-image matching (GIM) loss and a generative matching (GM) loss.

Inspired by Li et al. (2021), we firstly utilize a cross-modal encoder $f_c$ to fuse two unimodal representations and produce a joint representation of the graph-image pair and append a multilayer perceptron followed by softmax to predict the matching probability of $p^m = f_c(z_G, z_I)$. This can be formulated as:

$$\mathcal{L}_{GIM} = \mathbb{E}_{(G,I)\sim D}[\log P(y^m|(G, I))] \tag{6}$$

where $y^m$ is the ground-truth label representing whether the graph and the image are matched or not matched. The expected log-likelihood function could be defined as:

$$\mathcal{L}_{GIM} = -\frac{1}{P+N}\sum_{k=1}^{P+N}[y_k^m \cdot \log(p_k^m) + (1 - y_k^m) \cdot \log(1 - p_k^m)] \tag{7}$$

where $P$ denotes the number of positive pairs and $N$ denotes the number of negative pairs. In practice, $P : N = 1 : 1$. We use contrastive similarity (Eq. 3) to sample the in-batch *hard* negative pairs, where the molecular embeddings that are more similar to the image have a higher chance of being sampled (and vice versa). These pairs are hard to distinguish because there may exist other images different from the ground-truth that reflects the molecular perturbation equally well (or better) because of the batch effect Tran et al. (2020).

Motivated by recent success in generative contrastive learning Parmar et al. (2021); Liu et al. (2021a), we further employ variational auto-encoders (VAE) Kingma & Welling (2013) as generative agents

to reduce noise from experiments and enhance the cross-modal interaction among the hard negative pairs. In particular, the generative agents are asked to recover the representation of one modality given the parallel representation from the other modality. Herein, we performed cross-modal generation from two directions. This GM loss function can be formulated as:

$$\mathcal{L}_{GM} = -\frac{\lambda_{kl}}{2}(\mathcal{D}_{KL}(q_\phi(z_I|z_G)||p(z_I)) + \mathcal{D}_{KL}(q_\phi(z_G|z_I)||p(z_G)))$$
$$+ \frac{1}{2}(\mathbb{E}_{q_\phi(z_I|z_G)}[\log p_\theta(z_G|z_I)] + \mathbb{E}_{q_\phi(z_G|z_I)}[\log p_\theta(z_I|z_G)]) \tag{8}$$

where $\lambda_{kl}$ is the hyperparameter balancing the reconstruction loss. $p(z_I)$ and $p(z_G)$ is the prior of image embedding and graph embedding, respectively, and $q_\phi(z_G|z_I), q_\phi(z_I|z_G)$ is the corresponding posterior. Finally, the full contrastive objective of MIGA is:

$$\mathcal{L} = \mathcal{L}_{GIC} + \mathcal{L}_{MGM} + \mathcal{L}_{GIM} + \mathcal{L}_{GM} \tag{9}$$

## 4 EXPERIMENTS

### 4.1 DATASETS AND TASKS

**Pre-training Dataset.** We perform our experiments on the Cell Painting dataset CIL introduced by Bray et al. (2016; 2017). The dataset originally consists of 919,265 cellular images collected from 30,616 molecular interventions. Each image contains five color channels that capture the morphology of five cellular compartments: nucleus (DNA), Endoplasmic reticulum (ER), nucleolus/cytoplasmic RNA (RNA), F-actin cytoskeleton (AGP) and Mitochondria (Mito). A molecular intervention is photographed from multiple views in an experimental well and the experiment was repeated several times, resulting in an average of 30 views for each molecule. In order to keep the data balanced, we restricted each molecule to a maximum of 30 images and removed the untreated reference images, resulting in a cross-modal graph-image benchmark containing 750K views. We refer to this benchmark as CIL-750K. Detailed pre-process procedure and analysis of the pre-train dataset can be found in Appendix A.

**Downstream Tasks.** We evaluate the pre-trained model in three downstream tasks: graph-image retrieval, clinical outcome prediction and molecular property prediction.

- **Graph-Image Retrieval** contains two tasks: (1) graph as query and image as targets (Image retrieval); (2) image as query and graph as targets (graph retrieval/scaffold hopping). The goal of this task is to demonstrate whether the learned embeddings are able to preserve the inherent relations among corresponding graph-image pairs. We randomly split the CIL-750K dataset into a training set of 27.6K molecules corresponding to 680K images, and hold out the remaining data for testing. The held-out data consists of 3K molecules and the corresponding 50K images. We formulate the retrieval task as a ranking problem. In the inference phase, given a query molecular graph in the held-out set, we take images in the held-out set as a candidate pool and rank candidate images according to the L2 distance between the image embeddings and the molecular embeddings, and vice versa. The ranking of the ground-truth image/graph can then be used to calculate AUC, MRR (mean reciprocal rank), and Hit@1, 5, 10 (hit ratio with cut-off values of 1, 5, and 10). Experiments are performed 5 times with different seeds and the average results are reported.

- **Clinical Trial Outcome Prediction** aims to predict the clinical trial outcome (success or failure) of drug molecules. This task is extremely challenging as it requires external knowledge of system biology and trial risk from the clinical record. We use the Trial Outcome Prediction (TOP) benchmark constructed by Fu et al. (2022) for model evaluation. After dropping the combination mediation, the benchmark contains 944, 2865 and 1752 molecules for Phase I, II, and III tasks, respectively. We follow the data splitting proposed by Fu et al. (2022) and employ Precision-recall area under the curve (PR-AUC) and Area under the receiver operating characteristic curve (ROC-AUC) to measure the performance of all methods.

- **Molecular Property Prediction.** We further evaluate MIGA on six molecular property datasets: HIV, Tox21, BBBP,ToxCast, ESOL and Lipophilicity. These datasets are introduced by Wu et al. (2018) and further benchmarked by OGB communityHu et al. (2020a) for low-resource graph representation learning. Each data set contains thousands of molecular graphs as well as binary/scalar labels indicating the property of interest . We follow the OGB Hu et al. (2020a) setting and adopt the scaffold splitting strategy with a ratio for train/valid/test as 8:1:1 during fine-tuning.

| Task | Image Retrieval | | | | | Graph Retrieval | | | | |
|---|---|---|---|---|---|---|---|---|---|---|
| Metrics | MRR | AUC | Hit@1 | Hit@5 | Hit@10 | MRR | AUC | Hit@1 | Hit@5 | Hit@10 |
| Random Init | 0.051 | 0.500 | 0.010 | 0.050 | 0.100 | 0.051 | 0.500 | 0.010 | 0.050 | 0.100 |
| ECFP+CellProfiler | 0.052 | 0.511 | 0.010 | 0.049 | 0.100 | 0.053 | 0.500 | 0.011 | 0.050 | 0.100 |
| GraphCL | 0.247 | 0.818 | 0.106 | 0.391 | 0.558 | 0.279 | 0.835 | 0.132 | 0.433 | 0.616 |
| CLIP | 0.265 | 0.890 | 0.105 | 0.437 | 0.635 | 0.327 | 0.878 | 0.171 | 0.515 | 0.672 |
| CLOOB* | 0.278 | 0.895 | 0.137 | 0.451 | 0.612 | - | - | - | - | - |
| ALIGN | 0.288 | 0.821 | 0.148 | 0.434 | 0.594 | 0.379 | 0.810 | 0.214 | 0.581 | 0.713 |
| GIC | 0.288 | 0.847 | 0.148 | 0.434 | 0.594 | 0.280 | 0.907 | 0.125 | 0.447 | 0.652 |
| GIC+GIM | 0.303 | 0.876 | 0.145 | 0.485 | 0.660 | 0.337 | 0.927 | 0.176 | 0.523 | 0.693 |
| GIC+MGM | **0.409** | 0.913 | **0.244** | 0.612 | 0.741 | 0.405 | 0.931 | 0.244 | 0.598 | 0.737 |
| GIC+MGM+GIM | 0.401 | **0.926** | 0.230 | **0.616** | **0.743** | **0.433** | 0.935 | **0.275** | **0.628** | **0.739** |
| MIGA (Full) | **0.417** | **0.936** | **0.248** | **0.623** | **0.748** | 0.413 | **0.940** | 0.249 | 0.614 | **0.739** |

Table 1: Graph-image retrieval tasks on a held-out set of CIL-750K. MIGA and its variants are compared with adapted pre-training methods, GraphCL You et al. (2020), CLIP Radford et al. (2021), CLOOB Sanchez-Fernandez et al. (2022), ALIGN Jia et al. (2021), along with random initialization and a feature-based method ECFP Rogers & Hahn (2010) + CellProfiler Carpenter et al. (2006). *means re-implemented. The average of MRR, AUC, Hit@1, Hit@5 and Hit@10 are reported. The best and second best results are marked **bold** and **bold**, respectively.

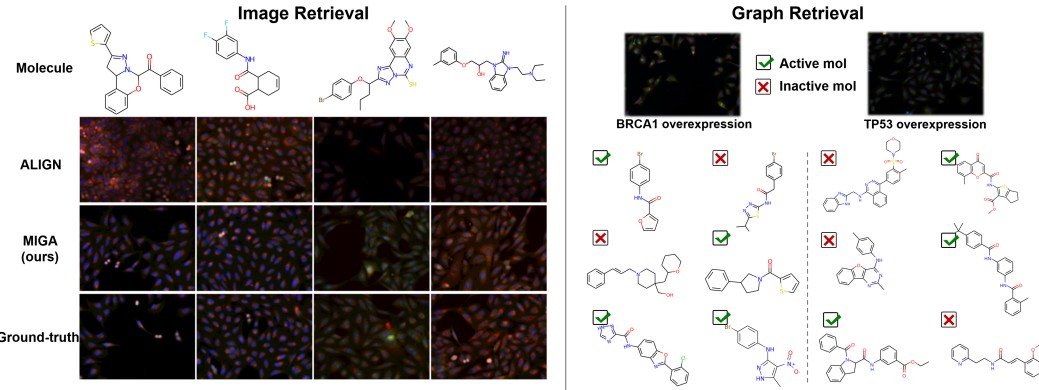

Figure 3: Case study on image retrieval task (left) and zero-shot image retrieval task (right). The images retrieved by our method and baseline method are shown on the left. Right side shows the cells induced by cDNA and our model can find diverse molecules that have similar functions to cDNA interventions (ticked). The full version is in Appendix D.

## 4.2 RESULTS

### 4.2.1 GRAPH-IMAGE RETRIEVAL

**Baselines.** Since we present improvements to pre-training models, we primarily compare our approach to the existing state-of-the-art graph SSL method, GraphCL You et al. (2020) and cross-modal pre-learning methods, CLIP Radford et al. (2021),CLOOB Sanchez-Fernandez et al. (2022) and ALIGN Jia et al. (2021). As none of them were specifically designed for graph-image cross modal learning, we adapted their encoders but followed their loss functions and tricks to perform pre-training on CIL-750K. We then evaluate the effectiveness of the proposed pre-training objectives (i.e., GIC, MGM, GIM, GGIM) with four variants of our method. For each baseline, we use the pre-trained model to output embeddings of molecular graphs and cellular images, then rank the candidate pool based on their similarity. We also include random initialization and feature engineering-based representation (ECFP Rogers & Hahn (2010) + CellProfiler Carpenter et al. (2006)) as baselines to demonstrate the challenge of the task. More details and hyperparameters have been attached in Appendix B.

**Result on CIL-750K held-out set.** Results of graph-image retrieval tasks are shown in Table 1. It is clear that our method MIGA significantly outperforms baseline methods including ECFP+CellProfiler, GraphCL, CLIP and ALIGN. For example, MIGA achieves 12.9% absolute MRR gain and 15.4% absolute Hit@10 gain over the best baseline ALIGN on the image retrieval task. These improvements

| Task | Phase I | | Phase II | | Phase III | |
|------|---------|---|----------|---|-----------|---|
| Metrics | PR-AUC | AUC | PR-AUC | AUC | PR-AUC | AUC |
| LR | 0.634 (0.007) | 0.487 (0.006) | 0.509 (0.014) | 0.534 (0.017) | 0.675 (0.010) | 0.528 (0.003) |
| RF | 0.651 (0.013) | 0.488 (0.009) | 0.488 (0.005) | 0.523 (0.009) | 0.722 (0.011) | 0.588 (0.013) |
| XGBoost | 0.646 (0.003) | 0.508 (0.006) | 0.481 (0.004) | 0.516 (0.007) | 0.712 (0.009) | 0.597 (0.015) |
| HINT | 0.683 (0.015) | 0.516 (0.005) | 0.537 (0.004) | 0.584 (0.003) | 0.689 (0.003) | 0.621 (0.006) |
| ContextPred | 0.693 (0.006) | 0.541 (0.019) | 0.544 (0.019) | 0.586 (0.003) | 0.710 (0.023) | 0.554 (0.036) |
| GraphLoG | 0.681 (0.016) | 0.539 (0.016) | 0.550 (0.043) | **0.593 (0.043)** | 0.719 (0.024) | 0.554 (0.024) |
| GROVER | 0.711 (0.015) | 0.559 (0.024) | 0.521 (0.005) | 0.574 (0.011) | 0.713 (0.013) | 0.575 (0.028) |
| GraphCL | 0.721 (0.020) | 0.578 (0.018) | 0.543 (0.008) | 0.588 (0.004) | **0.733 (0.011)** | **0.601 (0.008)** |
| JOAO | **0.736 (0.019)** | **0.586 (0.018)** | **0.546 (0.018)** | 0.587 (0.000) | 0.720 (0.000) | 0.563 (0.006) |
| MIGA | **0.758 (0.010)** | **0.601 (0.031)** | **0.562 (0.010)** | **0.605 (0.022)** | 0.729 (0.008) | **0.654 (0.016)** |

Table 2: Performance comparison of MIGA and several baselines for phase-level-outcome predictions on TOP dataset. We report the mean (and standard deviation) PR-AUC and ROC-AUC of five times. The best and second best results are marked **bold** and **bold**, respectively.

are consistent in the graph retrieval tasks, proving that the proposed method has superior performance in graph-image cross-modal learning. Compared to the basic loss (GIC), adding MGM and GIM both substantially improves the pre-trained model's performance across two tasks. This is in line with our expectation as GIC operates on unimodal representations while MGM and GIM operate on cross-modal representations. The proposed GGIM further enhances the model by reducing the noise and building explicit connections between two modalities.

**Case study for image retrieval.** We randomly select 20 pairs in the held-out set for case study. Figure 3 shows the results of four molecules, while the full version is attached in Appendix C. The result demonstrates that our model is more powerful in prioritizing the images that are similar to the ground-truth compared to ALIGN.

**Case study for zero-shot graph retrieval.** Of more practical value in the real world is the use of cellular images induced by non-small molecule interventions (e.g. cNDA Rohban et al. (2017)) to recall small molecules with similar effects. We collect 6 sets of cellular image induced by cDNA interventions for specific genes from Rohban et al. (2017). We use ExCAPEDB Sun et al. (2017) to retrieve active (agonists) and inactive molecules of these six genes as the candidate pools. Each gene has 20 active and 100 inactive molecules. Specifically, the mean Hit@10 for MIGA was $0.49 \pm 0.05$ , much higher than the random enrichment 0.167 (20/120), demonstrating the ability of our method to generalize to new datasets in a zero-shot manner. This result also indicates the practical potential of our method in virtual screening and drug repurposing. Figure 3 shows the results of two cellular images with cDNA-induced gene over-expression, the top-6 molecules ranked by MIGA are shown, see Appendix D for the full version.

### 4.2.2 CLINICAL TRIAL OUTCOME PREDICTION

**Baselines.** Since clinical prediction is a single-modality task, we compare the proposed method with the state-of-the-art SSL methods, ContextPred Hu et al. (2020b), GraphLoG Xu et al. (2021), GROVER Rong et al. (2020), GraphCL You et al. (2020) and JOAO You et al. (2021). We also included three machine learning-based methods (RF, LR, XGBoost) and a knowledge-aware GNN model HINT as listed in Fu et al. (2022).

**Results.** Results are summarized in Table 2. Our best model MIGA improves PR-AUC by an average of 1.67% and AUC by an average of 3.97% on the three clinical outcome prediction datasets, respectively, compared with the best SSL baseline JOAO. This is consistent with our hypothesis that phenotypic features can help predict tasks that involve complex biological processes.

### 4.2.3 MOLECULAR PROPERTY PREDICTION

**Baselines.** We compare the proposed method with the SOTA SSL methods including EdgePredHamilton et al. (2017), ContextPred Hu et al. (2020b), AttrMaskingHu et al. (2020b), GraphCL You et al. (2020), InfoGraph Sun et al. (2019), GROVERRong et al. (2020), GraphLoGXu

| Dataset | Classification (AUC) | | | | | Regression (RMSE) | | |
|---|---|---|---|---|---|---|---|---|
| | HIV | Tox21 | ToxCast | BBBP | Avg. | ESOL | Lipo | Avg. |
| Non-pretrain | 70.30 (0.51) | 68.90 (0.80) | 58.60 (1.20) | 65.40(2.4) | 65.80 | 1.278 (0.24) | 0.744 (0.14) | 1.011 |
| ContextPred | 74.17 (1.33) | 71.44 (0.11) | 60.05 (0.15) | 69.87 (0.99) | 68.88 | 1.141 (0.03) | 0.724 (0.02) | **0.933** |
| AttrMask | 75.55 (1.00) | 74.58 (0.66) | 59.51 (0.36) | 68.88 (2.65) | 69.63 | 1.194 (0.04) | 0.736 (0.02) | 0.965 |
| EdgePred | 72.53 (1.20) | 68.86 (0.38) | 57.39 (0.80) | 63.45 (1.39) | 65.56 | 1.146 (0.05) | 0.751 (0.02) | 0.949 |
| InfoGraph | **76.22 (0.24)** | 69.22 (0.78) | 59.87 (0.35) | 63.75 (1.52) | 67.27 | 1.242 (0.01) | 0.725 (0.01) | 0.984 |
| GraphLoG | 73.29 (2.64) | 69.80 (0.41) | 59.22 (1.05) | 68.43 (2.86) | 67.68 | 1.194 (0.02) | 0.766 (0.01) | 0.980 |
| GraphCL | 74.85 (1.71) | 74.19 (0.43) | 61.37 (0.10) | 66.13 (1.68) | 69.14 | 1.151 (0.04) | 0.745 (0.02) | 0.948 |
| GROVER | 74.35 (0.92) | 74.02 (0.79) | 61.30 (0.13) | **69.88 (0.58)** | **69.89** | 1.199 (0.02) | **0.721 (0.01)** | 0.960 |
| JOAO | 74.91 (0.66) | 74.60 (0.49) | **61.62 (0.37)** | 68.33 (0.58) | 69.87 | 1.117 (0.05) | 0.753 (0.02) | 0.935 |
| MIGA | **76.38 (0.55)** | 75.23 (0.71) | 62.34 (0.23) | 71.52 (0.43) | 71.37 | 1.123 (0.01) | **0.717 (0.00)** | 0.919 |

Table 3: Comparison of SSL baselines against MIGA on six OGB datasets. Mean ROC-AUC and Root Mean Squared Error (RMSE) (with the SD) of 5 times independent test are reported. The best and second best results are marked **bold** and **bold**, respectively.

| View | MRR | AUC | Hit@1 | Hit@10 |
|---|---|---|---|---|
| 1 | 0.336 | 0.927 | 0.175 | 0.692 |
| 5 | 0.419 | 0.933 | 0.251 | 0.728 |
| 10 | **0.427** | 0.938 | **0.263** | **0.742** |
| Full | 0.413 | **0.940** | 0.249 | 0.739 |

Table 4: Effect of # view per molecule.

| Methods | MRR | AUC | Hit@1 | Hit@10 |
|---|---|---|---|---|
| ResNet | **0.413** | 0.940 | **0.249** | 0.739 |
| EffciNet | 0.405 | **0.946** | 0.236 | **0.744** |
| DenseNet | 0.406 | 0.941 | 0.235 | 0.738 |
| ViT | 0.310 | 0.919 | 0.146 | 0.676 |

Table 5: Effect of CNN architecture.

et al. (2021), GraphCLYou et al. (2020), JOAOYou et al. (2021) . All the SSL models are pre-trained using released source code on CIL-750K data set. For fine-tuning, we follow the same setting with You et al. (2020; 2021).

**Results.** Results are summarized in Table 3. As we can see, MIGA performs the best on 5 out of 6 datasets and outperforms the existing SSL methods in the average performance by a large margin. The slightly lower performance in ESOL is expected as this property is related to the molecule's intrinsic characteristics, e.g., hydrophilicity and hydrophobicity. It does not rely on biological processes and phenotypic changes. Even though, MIGA will not cause negative transfer compared to non-pretrain baseline.

## 4.3 ABLATION STUDY

Table 4 shows the effect of the number of view per molecule perform on graph retrieval task. We notice that with less than 10 views, the more views involve in pre-training the better the results would be, but using all views (on average 25 views per molecule) does not achieve more gains. We attribute this phenomenon to the batch effect of the cellular images, where images from different wells can vary considerably because of the experimental errors and thus mislead the model.

Table 5 studies the impact of CNN architecture choices on the graph retrieval task. Due to the relatively small amount of data, we use small CNN variants (ResNet34He et al. (2016), EfficientNet-B1Tan & Le (2019), DenseNet121Huang et al. (2017) and ViT_tinyDosovitskiy et al. (2020)) for evaluation. We note that small CNN models such as ResNet, EfficientNet and DenseNet achieve comparable performance while bigger models like ViT do not show superiority. We assume this is because these models are pre-trained on non-cellular images, so heavier models do not necessarily lead to gains on cellular tasks.

## 5 CONCLUSIONS

In this paper, we propose a novel cross-modal graph pre-training framework, called MIGA. By introducing phenotypic features of cellur images during pre-training, MIGA is able to bring generalisable biological priors to graph neural networks, enabling them to be transferred to downstream biologically relevant tasks. Future directions include designing specific encoders for cellular images, optimizing cross-modal fusion mechanisms and extending the scale of the dataset to improve the performance of graph pre-training.

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
