# OpenReview forum: "Cross-modal Graph Contrastive Learning with Cellular Images"
_ICLR.cc/2023/Conference — Submitted to ICLR 2023_

### Official Review · Reviewer_PcWA · 2022-10-24

**Confidence:** 3
**Correctness:** 3
**Technical Novelty And Significance:** 2
**Empirical Novelty And Significance:** Not applicable
**Recommendation:** 5

**Clarity, Quality, Novelty And Reproducibility:**

Regarding clarity:
The paper is well written in general and can be improved by filling in some missing details:
- Parameters to generate masked context graph modeling in eq 4 and 5. It would also be helpful to clarify the cross-entropy used in eq 5.
- Eq 7: There should be k=1 instead of k in the summation?
- Details about the VAE architecture and training in Generative Graph-image matching are not provided.
- It would be helpful to discuss what features other than molecular embeddings are used in Clinical Outcome Prediction and how it fits into the pre-trained GNN.
 - It would be helpful, if possible, to provide details of cDNA and molecules used in the case study of zero-shot graph retrieval. Would be good to see Hit@10 of other SOTA as well.

Regarding quality and originality:
The work is a nice application of representation learning in biology and drug discovery, albeit It seems the paper did not provide a new technical contribution. The idea of contrastive learning from chemical structures and cell images seems not new.  Performance improvement is properly demonstrated by evaluation experiments and case studies.


**Strength And Weaknesses:**

Strength:
- The work tackles the important problem of learning molecular representations by bringing high-throughput imaging data.
- Performance improvement of the pre-training strategy is demonstrated on three different downstream tasks compared with SOTA.
- An interesting case study is reported to find molecules that have similar effects as single gene overexpression.

Weakness
- While I am no expert on cross-modal learning, it seems the framework just embraces several state-of-the-art ideas in contrastive learning without novel technical contributions.
- No ablation studies for combinations of contrastive losses
- Given the application nature of this work, related work should discuss the state of the arts in learning embedding from cell images. Particularly, the published workSanchez-Fernandez, Ana, et al. "Contrastive learning of image-and structure-based representations in drug discovery." ICLR2022 Machine Learning for Drug Discovery. 2022.
seems conceptually similar to the proposed work.


**Summary Of The Paper:**

The paper presents a cross-model framework to learn molecular embeddings jointly from the chemical structure and images of drug-treated cells. Using four contrastive losses to pre-train a model on a high-throughput imaging dataset, the paper demonstrated improved performance on three downstream tasks, including retrieving chemical structures or images from the other channel, predicting clinical outcomes, and predicting molecular properties.


**Summary Of The Review:**

I would not recommend this paper for publication in ICLR due to my concern about the limited novelty and tech contribution. It is a nice piece of work that I enjoyed reading through.

---

> ### Author Response · Authors · 2022-11-16
> **Response to Reviewer PcWA**
>
> We thank the reviewer for the time taken to review our work and the constructive feedback provided. We have addressed the reviewer's specific questions below.
>
> > Q1. While I am no expert on cross-modal learning, it seems the framework just embraces several state-of-the-art ideas in contrastive learning without novel technical contributions.
>
> Response: Thanks for the valuable comments. We respectfully disagree and want to highlight the methodological novelties in the design of two complementary cross-modal loss functions (MGM and GGIM) for two heterogeneous modalities. Specifically,
>
> a) The MGM loss aims to learn the local molecular graph patterns with the help of both the corresponding image embedding and the partial molecular graph. To our knowledge, this is the first graph-image cross-modal masking prediction module.
>
> b) GGIM is used to distinguish further the hard negative graph-image pairs that share similar global semantics but differ in fine-grained details. Specifically, the matching loss of GGIM samples hard negative pairs from the same batch following the contrastive similarity distribution calculated by GIC (i.e., the molecular embeddings that are more similar to the image have a higher chance to be sampled, and vice versa), to further learn the difference between them and positive pair. To our knowledge, the combination of GM and GGIM for progressive information fusion is also novel in contrastive learning.
>
> > Q2. No ablation studies for combinations of contrastive losses.
>
> Response: We did show the combinations of contrastive losses in Table 1, including GIC+ GIM, GIC+MGM, etc. We did not show MGM+GIM combo because GIM cannot be trained on its own without GIC. Note that GIM is used to distinguish the hard negative graph-image pairs calculated by GIC.
>
> > Q3. Related work should discuss the state of the arts in learning embedding from cell images.
>
> Response: Thanks for your valuable suggestion. Briefly, CLOOME employs a CLIP-like loss to pre-train the cellular image encoder with molecular fingerprints. Different from their focus on the image encoder, MIGA aims at extracting information from cellular images to make enhanced molecular representations and designed two complementary cross-modal contrastive losses for information interaction in an end-to-end manner. We have stated the difference with [1] in the comments to all reviewers and have added this in the revised paper.
>
> > Q4. Clarity
>
> Response: We appreciate the reviewer's careful review of our paper. We have corrected the typos in the revised version and added some descriptions in response to the reviewer’s concern.
> -   Parameters to generate masked context graph modeling; tpyo of eq 7
>
> Response: We randomly mask the atom/bond attributes and constructed the context graphs to generate the masked context graph without parameters in eq 4 and 5. We have clarified the cross-entropy in eq 5 and k=1 in the summation of eq 7 in the revised paper.
>
> - Details about the VAE architecture and training in Generative Graph-image matching are not provided.
>
> Response: Details about the VAE architecture and training in Generative Graph-image matching have been added to Appendix accordingly.
>
> - It would be helpful to discuss what features other than molecular embeddings are used in Clinical Outcome Prediction and how it fits into the pre-trained GNN.
>
> Response: The core of our work is to demonstrate that the information we extract from the cell images is useful for the pre-trained GNN and the results in Table 3 have verified the statement. The other features, if needed, could be fit into the pre-trained GNN on the last classification layer, but does not help to elucidate that conclusion.
>
> - It would be helpful, if possible, to provide details of cDNA
>
> Response: The details of cDNA and molecules used in the case study of zero-shot graph retrieval have been included in Appendix. As this is a new application scenario that we built, there has been no previous work on it. All the molecules would be provided after acceptance.
>
> We appreciate the reviewer's valuable suggestions and hope that the new discussion and results increase the reviewer's confidence in the significance of our work. We are happy to answer any additional questions.
>
> [1] Sanchez-Fernandez, A., Rumetshofer, E., Hochreiter, S., & Klambauer, G. (2022, March). Contrastive learning of image-and structure-based representations in drug discovery. In ICLR2022 Machine Learning for Drug Discovery.

---

> > ### Author Response · Authors · 2022-11-27
> > **Thank you & Looking forward to reply**
> >
> > Dear Reviewer PcWA:
> >
> > Thank you very much for your time and constructive comments. According to your reviews, we responded with detailed explanations and revised the manuscript, which we believe have covered your concerns. Please let us know if you still have any unclear points in our work. We are happy to discuss this further.
> >
> > Best,
> >
> > Authors

---

> > > ### Comment · Reviewer_PcWA · 2022-11-28
> > > **Thanks for the detailed response**
> > >
> > > Thanks for the detailed response. I don't think I am fully convinced about the issue of the novelty of the paper. I update my score to 5.

---

> > > > ### Author Response · Authors · 2022-11-28
> > > > **Thank you**
> > > >
> > > > Dear reviewer,
> > > >
> > > > Thank you for raising the score. We appreciate your engagement and reviewing service. Please let us know if there are items that could help improve your rating. Thanks!
> > > >
> > > > Best regards

---

### Official Review · Reviewer_4Jez · 2022-10-25

**Confidence:** 5
**Correctness:** 2
**Technical Novelty And Significance:** 1
**Empirical Novelty And Significance:** 2
**Recommendation:** 3

**Clarity, Quality, Novelty And Reproducibility:**

### Clarity
The paper is clearly written and structured.

### Novelty
The almost exact same approach has been presented in ref [1]. Even the identical dataset (cell painting) and contrastive learning objectives (InfoNCE, CLIP-like) have been used. The authors should propose a new method, approach and concept and state their novel contributions, e.g., at the end of the Introduction section.

### Quality
i) There is a fundamental problem with the approach and this is the focus on pre-training the graph neural network, i.e. molecule encoder, using cellular images. Many changes in molecular structure, do not lead to changes in cell morphology, such that the graph network is not required to code these changes in the learned representation. However, for downstream tasks exactly these structural changes might be relevant. Comparing to CLIP: CLIP focuses on pre-training the image-encoder (equivalent here: cellular image encoder) and not on the text-encoder (equivalent here: graph encoder) because for the text-encoder other pre-training tasks, such as masking/de-noising, are more appropriate. The authors should justify their focus on the graph neural network encoder.

ii) It is unclear why multiple types of losses are necessary to train this architecture. For CLIP-like methods, the loss in Eq(3) is usually sufficient. Since the loss terms (Eq (9)) are not balanced against each other (except for Eq(8)), it is very likely that one of the loss terms dominates over the others, such that the training could completely neglect the image-modality. The ablation study (Section 4.3) does not investigate this at all. However, the bottom of Table 1 provides some insights: the strongest increase of performance arises from the masking loss, indicating that the cellular images are hardly used. Therefore, a pre-training strategy purely using the MGM loss (Eq 4,5) might be superior to cross-modal pretraining. Due to the missing error bars, it is unclear whether there is a difference between MIGA and “GIC+MGM” at all. The authors should justify their design choices and perform an ablation study to demonstrate that all four loss terms are necessary and need not be balanced against each other.

iii) Missing embedding into related work. The author's are unaware of closely related work ([1]) and also unaware that their suggested graph-image contrastive loss (Eq (3)) is identical to the CLIP algorithm. The authors focus on pre-training the graph neural network to encode molecules, but are unaware of related work on molecule encoders that perform much better on the MoleculeNet tasks (Table 3).

iv) Informative baselines are missing. I quickly compared against a simplistic baseline, a fixed encoder (ECFP [7]) and linear probing and reached 77.05 on Tox21 and 69.43±0.0 on ToxCast. This baseline outperforms MIGA indicating that the graph neural network has not even learned to encode as well as an encoder that just checks for all substructures [7]. Furthermore, the best methods in the Tox21 data challenge reached an AUC of 0.845, with mostly descriptor-based approaches at that time, which means that the pre-training task for the GNNs is lacking. Comparing how much computational effort (energy and carbondioxide costs) it takes to pre-train the MIGA molecule encoder that in the end does not even perform better than an encoder that costs almost nothing (ECFP), makes it really difficult to justify the use of MIGA. Or in other words: the suggested approach to properly pre-train graph neural networks would mean that first a multi-million dollar experiment to collect the microscopy images has to be done. The authors should include informative and state-of-the-art baselines into their method comparisons. In table 2, it is unclear on which representation LR, RF and XGboost are running. The authors should use standard molecular fingerprints and descriptors as baselines in all studies.

v) Presented performance metrics in many of the tables (e.g. Table 3 and 4) are results of single runs and therefore any performance differences can arise just by chance. For the ones with standard deviations it is unclear what they represent: re-runs on the same split or cross-validation? The authors should perform re-runs, or cross-validation, to obtain error bars and confidence intervals on all performance metrics.

### Reproducibility
The dataset is publicly available, but needs strong pre-processing to be used for ML. The authors provide code as supplementary material, which makes the work likely to reproduce. One difficulty could be the dependency on the preprocessing of the Cellpainting dataset

References:
[1] Sanchez-Fernandez, A., Rumetshofer, E., Hochreiter, S., & Klambauer, G. (2022, March). Contrastive learning of image-and structure-based representations in drug discovery. In ICLR2022 Machine Learning for Drug Discovery.
[2] Yang, K., Swanson, K., Jin, W., Coley, C., Eiden, P., Gao, H., ... & Barzilay, R. (2019). Analyzing learned molecular representations for property prediction. Journal of chemical information and modeling, 59(8), 3370-3388.
[3] Ma, J., Sheridan, R. P., Liaw, A., Dahl, G. E., & Svetnik, V. (2015). Deep neural nets as a method for quantitative structure–activity relationships. Journal of chemical information and modeling, 55(2), 263-274.
[4] Jiang, D., Wu, Z., Hsieh, C. Y., Chen, G., Liao, B., Wang, Z., ... & Hou, T. (2021). Could graph neural networks learn better molecular representation for drug discovery? A comparison study of descriptor-based and graph-based models. Journal of cheminformatics, 13(1), 1-23.
[5] Wang, H., Kaddour, J., Liu, S., Tang, J., Kusner, M., Lasenby, J., & Liu, Q. (2022). Evaluating Self-Supervised Learning for Molecular Graph Embeddings. arXiv preprint arXiv:2206.08005.
[6] You, Y., Chen, T., Shen, Y., & Wang, Z. (2021, July). Graph contrastive learning automated. In International Conference on Machine Learning (pp. 12121-12132). PMLR.
[7] Rogers, D., & Hahn, M. (2010). Extended-connectivity fingerprints. Journal of chemical information and modeling, 50(5), 742-754.


**Strength And Weaknesses:**

Strengths:
- A relevant problem is approached. The retrieval-task between cellular images and molecular graphs is appealing and significant.
- The paper is clearly written and structured; Figures are informative; Notation follows ML standards
- The authors took care of splitting the data appropriately since multiple  images correspond to one molecular graph

Weaknesses:
- The approach is not novel, but has been suggested before.
- The presented results are not relevant due to missing embedding into related work, missing informative baselines and incorrect scope.
- The presented work has several severe technical errors. There is also a fundamental problem in the approach (see below).

**Summary Of The Paper:**

The authors propose an approach to co-learn representations from molecular graphs and fluorescence microscopy images.

**Summary Of The Review:**

The main idea of the work is not novel, there is a fundamental problem with the approach, i.e. that microscopy images cannot capture all changes in molecular structure, and the experiments are insufficient due to lacking baselines. The choice of the loss function with many terms is particularly problematic for learning, but this is not ablated.

---

> ### Author Response · Authors · 2022-11-16
> **Response to Reviewer 4Jez (Part 1 of 3)**
>
> We thank the reviewer for their valuable comments and suggestions. Briefly, the reviewer has two main concerns: (1) Our method is similar to CLOMME but has a “wrong” focus on the graph encoder, and (2) the designed losses are unnecessary to train this architecture. With all due respect, we disagree on both points and argue that the reviewer may misunderstand our work. By adding extra experiments and explanations, we hope our rebuttal can clarify the reviewer’s concerns.
>
> > Q1: The approach is not novel, but has been suggested before; should state their novel contributions.
>
> Response: Thanks for the valuable suggestions. We apologized for the unclear statement of novelty and differentiated the CLOOME with MIGA in the revised paper. The details of the differences can also be found in the comment to all reviewers. The novelty of paper can be summarized as follows:
> 1. In contrast to the insignificant gains in biologically relevant properties achieved by previous uni-modal molecular graph pre-training efforts, this paper comprehensively utilizes high-content cell microscopy images to assist in learning molecular representation. This can help researchers to re-think new cross-modal pre-training rather than uni-modal pre-training.
> 2. The paper constructs two novel cross-modal benchmarks and proposes a novel cross-modal training framework with multiple adaptive loss functions to achieve SOTA performance on each benchmark. In particular, we are far ahead of the SOTA methods (both supervised and unsupervised learning) in clinical outcome predictions, demonstrating that the introduction of cellular images is beneficial for tasks involved in intricate biological processes. This also implies that our model can be used by practitioners.
> 3. The zero-shot experiments on cDNA images show the potential of the approach to be applied to real-world drug discovery without any specialization to particular cellular images. This is an important point of departure from prior works [1][2][3], as current molecular graph pre-training methods require fine-tuning specific downstream datasets.
> We will list these in the final version of the paper once more space were given.
>
> > Q2: There is a fundamental problem with the approach and this is the focus on pre-training the graph neural network, i.e. molecule encoder, using cellular images...However, for downstream tasks exactly these structural changes might be relevant.
>
> Response: We have different idea with the suggestion of focusing on pre-training the image-encoder only. The specific focus on which encoder is relevant for what the problem is to be solved. As repeatedly highlighted in the text, we expect the model to help molecular representations learn beyond structure, thus helping downstream molecular-relevant tasks rather than image-related ones. Data that do not lead to changes in cell morphology can also tell the network that certain molecules have properties that are not responsive to that cell line, which is also important side information beyond chemical structure. Furthermore, considering the specific batch effect of the cellular images and our counter-intuitive ablation results on the image encoders, it is not appropriate to simply equate this task with text-image pre-training. Note that masking and de-noising are not suitable for graph pre-training as they will cause property cliffs for small molecules. This is also the reason why we introduce a heterogeneous modality for molecular graph pre-training.
>
> > Q3-1: It is unclear why multiple types of losses are necessary to train this architecture. It is very likely that one of the loss terms dominates over the others, such that the training could completely neglect the image modality. However, the bottom of Table 1 provides some insights: the strongest increase in performance arises from the masking loss, indicating that the cellular images are hardly used.
>
> Response: This is a misunderstanding. Note that MGM is a customized cross-modal masking loss that learns the local molecular graph patterns with the help of **both the corresponding cellular image embedding and the partial molecular graph**. This is novel and is completely different from the single-modality masking loss. Given both GIC, MGM and GGIM have cross-modal interaction, it is **impossible** that the training process completely neglects the image modality.

---

> > ### Author Response · Authors · 2022-11-16
> > **Response to Reviewer 4Jez ( Part 2 of 3)**
> >
> > > Q3-2: Due to the missing error bars, it is unclear whether there is a difference between MIGA and “GIC+MGM” at all. The authors should justify their design choices and perform an ablation study to demonstrate that all four loss terms are necessary and need not be balanced against each other.
> >
> > Response: Thanks for the suggestion. We did not report the missing error bars of Table 1 because of the stable results and limited space. To address your concern, we added the SD of the results and show the MGM independently.
> > | Task| 	MRR| 	AUC| 	Hit@1| 	Hit@5	| 	Hit@10|
> > | ------ | ------ | ------ | ------ | ------ | ------ |
> > | GIC | 0.288 (0.002) |  0.847(0.005) |  0.148(0.003) |  0.434(0.003) | 0.594(0.006)|
> > | MGM | 0.247(0.003) | 0.818(0.008) | 0.106(0.005) | 0.391(0.008)|  0.558(0.007)|
> > | GIC+MGM |0.409(0.004)|  0.913(0.004)|  0.244 (0.004)|  0.612(0.003) |  0.741(0.006)|
> > | MIGA (Full)| 0.417 (0.005)|0.936 (0.006) |0.248 (0.005) | 0.623 (0.004)| 0.748 (0.006)|
> >
> > From the table, we can observe that MGM cannot perform well when trained alone. MIGA, combining GIC, MGM and GGIM together, performs the best. We note that GGIM cannot be independently trained as it is used to distinguish the hard negative graph-image pairs calculated by GIC, which means the pairs that share similar global semantics but differ in fine-grained details. This has been highlighted in red in the revised paper. We also want to emphasize that MGM and GGIM are designed to complement the GIC and GIC is an important cornerstone of cross-modal contrastive learning. The difference and correlation between these three losses have been repeatedly justified in the submission.
> >
> > > Q4: Missing embedding into related work of CLOOME, CLIP and MolecularNet.
> >
> > Response: Thanks for the constructive comment. We agree and thus add the suggested workshop paper, CLOOME [1], for comparison. Although no code has been released yet, we re-implement it as a baseline following the settings of the paper. Here we show the results of the graph retrieval task (the only overlap experiments):
> > |Task|MRR	|AUC	|Hit@1	|Hit@5		|Hit@10|
> > | ------ | ------ | ------ | ------ | ------ | ------ |
> > |ECFP+CellProfiler	|0.052	|0.511	|0.010	|0.049	|	0.100|
> > |CLIP|	0.265	|0.890	|0.105	|0.437	|	0.635|
> > |CLOOME*	|0.278	|0.895	|0.137	|0.451	|	0.612|
> > |**MIGA** 	|**0.417**	|**0.936**|**0.248**	|**0.623**		|**0.748**|
> >
> > We find that this state-of-the-art solution does not consistently provide improvements and is significantly outperformed by our MIGA method. This is expected as the CLOOME is a CLIP-like method, while MIGA adapted extra cross-modal infusion modules for augmentation. We also update the tables in the paper. Please refer to Section 4.2.1 and Appendix for more details.
> >
> > Note that the InfoNCE (Eq(3)) loss is employed by CLIP but not created by CLIP. This is currently one of the most popular losses in contrastive works and we have already referred to CLIP clearly in related work and baseline methods. We have also already referred to MolecularNet. We don’t compare to the molecule encoders because our results were based on **OGB scaffold split **(https://ogb.stanford.edu/) while the results in MoleculeNet are based on **random split**.
> >
> > > Q5: Informative baselines (supervised learning) in molecular properties predictions are missing.
> >
> > Response: With all due respect, we disagree with the suggestion of introducing extra supervised baselines. This involves some basic understanding of the recent progress of self-supervised learning in molecular graphs and again reveals that the reviewer misunderstood our work. We listed the main points below.
> >
> > Briefly, the key motivation of the self-supervised method in molecular graphs is to see **if the pre-training framework can extract the side information from the unlabelled data effectively**. Thus, the core to be compared is a basic GNN (usually GIN[4]) with **random initialization**. We have compared all these SOTA pre-training methods following the same pre-training and fine-tuning settings[5][6][7][8]. We also mentioned these settings in the main text and Appendix B yet unfortunately missed by the reviewer.
> > In the case of clinical outcome prediction, HINT[49] is a hybrid method that utilizes advanced MPNN with augmented chemical features and shows superior to the original MPNN and DNN methods in the original paper. We re-implemented HINT in our datasets and demonstrated that MIGA outperformed it in the clinical prediction tasks by a large margin.
> > In summary, we do not see the need to add extra supervised baselines, as our motivation is to prove that our framework is able to distill information from cellular images effectively.

---

> > > ### Author Response · Authors · 2022-11-16
> > > **Response to Reviewer 4Jez ( Part 3 of 3)**
> > >
> > > > Q6: Presented performance metrics in many of the tables (e.g. Table 3 and 4) are results of single runs and therefore any performance differences can arise just by chance.
> > >
> > > Response: Thanks for the suggestions. For all the experiments, the reported values are already averaged over five runs as we mentioned in the appendix. We stated them more clearly in the revised paper. Tables 2 and 3 have already shown the standard deviation. We did not put them into tables 1 and 4 because of the stable outcomes and limited space.
> > >
> > > Summary
> > > We appreciate the reviewer's valuable suggestion and hope that the new discussion and results increase the reviewer's confidence in the significance of our work. We are happy to answer any additional questions.
> > >
> > > [1] Sanchez-Fernandez, A., Rumetshofer, E., Hochreiter, S., & Klambauer, G. (2022, March). Contrastive learning of image-and structure-based representations in drug discovery. In ICLR2022 Machine Learning for Drug Discovery.
> > >
> > > [2] Teru K, Denis E, Hamilton W. Inductive relation prediction by subgraph reasoning, International Conference on Machine Learning, 2020.
> > >
> > > [3]Yang F, Yang Z, Cohen W W. Differentiable learning of logical rules for knowledge base reasoning. Advances in neural information processing systems 2017.
> > >
> > > [4] Keyulu Xu, Weihua Hu, Jure Leskovec and Stefanie Jegelka. How powerful are graph neural networks? ICLR 2019.
> > >
> > > [5] Weihua Hu, Bowen Liu, Joseph Gomes, Marinka Zitnik, Percy Liang, Vijay S. Pande, and Jure Leskovec. Strategies for pre-training graph neural networks. ICLR 2020.
> > >
> > > [6] You, Yuning and Chen, Tianlong and Sui, Yongduo and Chen, Ting and Wang, Zhangyang and Shen, Yang. Graph contrastive learning with augmentations. NeurIPS 2020.
> > >
> > > [7] Minghao Xu, Hang Wang, Bingbing Ni, Hongyu Guo, and Jian Tang. Self-supervised graph-level representation learning with local and global neural networks. ICML 2021.
> > >
> > > [8] Shengchao Liu, Hanchen Wang, Weiyang Liu, Joan Lasenby, Hongyu Guo, and Jian Tang. Pre-training molecular graph representation with 3d geometry.ICLR 2022.

---

> ### Author Response · Authors · 2022-11-27
> **Looking forward to reply**
>
> Dear Reviewer 4Jez:
>
> Thank you very much for your precious time and valuable comments. According to your reviews, we responded with detailed explanations and new experimental results, which we believe have covered your concerns. Please let us know if you still have any unclear points in our work. We are happy to discuss this further.
>
> Best,
>
> Authors

---

### Official Review · Reviewer_pEPS · 2022-10-28

**Confidence:** 5
**Clarity, Quality, Novelty And Reproducibility:** The paper is of high quality and is c…
**Correctness:** 4
**Technical Novelty And Significance:** 3
**Empirical Novelty And Significance:** 3
**Recommendation:** 8

**Strength And Weaknesses:**

1- The proposed method, MIGA, utilizes the cross-modal information that exist naturally in microscopy images to improve graph data representations. While the underlying algorithms and losses to perform this cross-modal alignment are not novel, the idea of employing these heterogeneous modalities is inspired and novel.
2- The paper is easy to understand, and the used language is clear.
3- The experimental results are comprehensive, and confirm the merits of the proposed method. Particularly remarkable are the improvements obtained on clinical outcome prediction tasks.

**Summary Of The Paper:**

This work utilizes the self-supervised learning paradigm to improve the data representations in Graph Neural Networks. Different from previous literature, the proposed method employs cross-modal correspondences between the Graphs of molecular structures and the associated (paired) cell microscopy images. The intuition behind combining these specific modalities is clear, drug molecules have the ability to cause perturbations (subtle morphological changes) to cells, e.g. their shapes, numbers, structures, and so on. To that end, the paper proposes a pretraining method called MIGA that processes these paired modalities, and aligns them in the feature space. MIGA is trained with three combined loss terms, which are inspired from existing literature on contrastive learning and generative modeling. Then, the learned data representations, stored in the form of neural network weights in the Graph and Image encoders, are evaluated on several downstream tasks. The chosen downstream tasks seem relevant to the domain, and are sensible overall. The reported results on these downstream tasks confirm the improvements obtained by pretraining by their method, MIGA, as opposed to the baselines from literature.

**Summary Of The Review:**

Overall, I recommend accepting the paper. The method is novel, the writing is clear, and the experimental results are sound.

---

> ### Author Response · Authors · 2022-11-16
> **Response to Reviewer pEPS**
>
> We appreciate the reviewer's careful review of our paper, positive feedback and constructive comments.

---

### Official Review · Reviewer_mr25 · 2022-10-31

**Confidence:** 2
**Correctness:** 4
**Technical Novelty And Significance:** 4
**Empirical Novelty And Significance:** 4
**Recommendation:** 6

**Clarity, Quality, Novelty And Reproducibility:**

Clear:
- Overall, the paper is well-written and easy to follow. The logic of the introduction, methods, and results flow well.
- The figures give helpful explanation of the context of the problem and the method and its applications.

Unclear:
- The cell images are in general very small throughout the paper. It is hard to perceive the details. For example, in the examples for image retrieval in Fig. 3, the individual cells are very small. The images of molecules are also small in these figures, though this is not as significant of an issue as the cell images.
-  A related issue is the contrast of the images being very low. Even after zooming in, I admit I cannot perceive much difference between the images of BRCA1 overexpression and TP53 overexpression.

- The method seems reproducible.
- The method is significantly novel, to my knowledge.

**Strength And Weaknesses:**

Strengths:
- The method is well-reasoned and is intriguing in this context, in what is a very challenging problem but of great relevance for the biomedical community.
- MIGA is applicable to several important real-world applications, namely clinical trial outcome and molecular property prediction.
- The authors compare to an impressive number of baselines, showing a consistent advantage of their method.
- Related, the authors have thoroughly considered proper metrics and ablation studies to evaluate MIGA.

Weaknesses:
- I am concerned that there may be other works in this area that are not considered. There are entire companies, such as Recursion, that work in this area of deep learning-based high-content image screens for drug design. I do not have a sense of how significant of an advance is their method in comparison to such work.
- The performance increase is modest, which is expected given the difficulty of the problem. I believe it is worthwhile to publish still, given the consistency of the improvement for such an important problem, but it is still a weakness that it does not yield a more significant improvement.

**Summary Of The Paper:**

The authors present MIGA, a method for combining graphs of molecules and corresponding high-content cell images into a related embedding space. This learned embedding space can then be used for several tasks of clinical relevance, namely cell image retrieval, predicting clinical outcomes of drugs, and molecular property prediction. The authors evaluate MIGA against several reasonable baselines in all areas.

**Summary Of The Review:**

Overall, the paper is well-done and the approach is an intriguing way to combine cell images and molecular structure. The experiments are very well-done and thought out. My hesitations about giving a more confident recommendation to accept are that the improvements, while consistent, are modest, and I am concerned that there is much other work out there that I am not aware of, given how many companies and researchers are working in this area.

---

> ### Author Response · Authors · 2022-11-16
> **Response to Reviewer mr25**
>
> We thank the reviewer for the valuable suggestions and questions! As for the comments:
>
> > Q1. I am concerned that there may be other works in this area that are not considered. I do not have a sense of how significant of an advance is their method in comparison to such work.
>
> Response: Thanks for the constructive comment! We agree and thus add a recent workshop paper, CLOOME [1], for comparison, as also suggested by other reviewers. Although no code has been released yet, we re-implement it as a baseline following the settings of the paper. Here we show the results of the graph retrieval task (the only similar task with CLOOME):
>
> |Task|MRR	|AUC	|Hit@1	|Hit@5		|Hit@10|
> | ------ | ------ | ------ | ------ | ------ | ------ |
> |ECFP+CellProfiler	|0.052	|0.511	|0.010	|0.049	|	0.100|
> |CLIP|	0.265	|0.890	|0.105	|0.437	|	0.635|
> |CLOOME*	|0.278	|0.895	|0.137	|0.451	|	0.612|
> |**MIGA** 	|**0.417**	|**0.936**|**0.248**	|**0.623**		|**0.748**|
>
> We find that this state-of-the-art solution marginally improves CLIP and is significantly outperformed by our MIGA method. This is expected as the CLOOME is a CLIP-like method, while MIGA adapted more cross-modal infusion modules for augmentation. We also update the tables in the paper. Please refer to Section 4.2.1 and the Appendix for more details.
>
> > Q2. It is still a weakness that it does not yield a more significant improvement.
>
> Response: Thanks for the valuable comment. We partially disagree with the statement because Table 1 has shown that our method gains significant improvements over CLOOME (13.9% absolute MRR gains and 13.6% absolute Hit@10 gains). The graph-image retrieval is the most direct task that can reflect the information we learned from the data and is also the key proof of concept. And the benefits are consistent in clinical prediction tasks, where we also gained remarkable improvements compared to baseline methods. In terms of the modest improvement in molecular property predictions, we argue that it is because the properties selected are not all correlated to the images, as also highlighted in red in the revised section 4.2.3. Further, the small amount of training data available is also an important factor, and we believe that the problem will be solved as the number of data increases (like recently released JUMP datasets [2]). We have added these in the revised paper.
>
> > Q3. Contrast and quality of the images.
>
> Response: Thanks for the valuable suggestions! We realize the problem but currently cannot change too much given the limited space of the format. We will improve the composition and the quality of the images in the final version.
>
> We appreciate the reviewer's valuable suggestions and hope that the new results and explanation increase the reviewer's confidence in the significance of our work.
>
> [1] Sanchez-Fernandez, A., Rumetshofer, E., Hochreiter, S., & Klambauer, G. (2022, March). Contrastive learning of image-and structure-based representations in drug discovery. In ICLR2022 Machine Learning for Drug Discovery.
>
> [2] https://jump-cellpainting.broadinstitute.org/results

---

> ### Author Response · Authors · 2022-11-27
> **Looking forward to reply**
>
> Dear Reviewer mr25:
>
> Thank you very much for your valuable time and constructive comments. According to your reviews, we responded with detailed explanations and additional experiments, which we believe have covered your concerns. Please let us know if you still have any unclear points in our work. We are happy to discuss this further.
>
> Best,
>
> Authors

---

### Author Response · Authors · 2022-11-16
**Comments to all reviewers.**

We thank all reviewers for their insightful comments and suggestions. We are glad that many of the reviewers found this to be a novel and interesting work, for example, "the method is significantly novel" (mr25) "the idea of employing these heterogeneous modalities is inspired and novel." (GR4V) and "the retrieval-task between cellular images and molecular graphs is appealing and significant." (4Jez). Some reviewers raise concerns about the differences from a recent workshop paper [1], named CLOOME. We carefully read this paper and found these two works share similar intuitions on a high level but have many fundamental differences. The major ones are summarized as follows:

1. **Graph pre-training VS image pre-training**: CLOOME pre-trains the image encoder with non-differentiable molecular fingerprints, while MIGA aims at pre-training both the graph and image encoders simultaneously but emphasizes on pre-trained graph encoder for molecular representation. We expect the model to help molecular representations learn beyond structure, thus helping downstream molecular-relevant tasks rather than image-related ones. This is more generalizable as molecular graph structures are much easier to be accessed than high-content images. In contrast, CLOOME focuses on learning image encoders for enhancing cellular imaging representations.

2. **Different application scenarios**: Given the advantages in graph and image encoders co-training, MIGA can naturally be applied to various tasks from both sides, such as graph-image mutual retrieval, molecular properties predictions, and clinical outcome predictions. In contrast, the CLOOME focuses on image-side tasks like graph retrieval and image-based predictions.

3. **Cross-modal fusion**: During the contrastive learning process, CLOOME employs single CLIP-like loss for optimization and because of the non-differentiable nature of molecular fingerprints, the model has difficulties in performing cross-modal information fusion. Instead, MIGA adapted three advanced contrastive losses, and two of them (MGM and GIM) are customized for cross-modal information interactions between molecular graph encoders and cellular image encoders. Moreover, the whole framework is trained in an end2end fashion.

To make the contribution clear, we added the re-implemented CLOOME (as no code has been released) as a baseline method and discussed it in the updated paper. Also, we integrated every piece of feedback in the new version of the paper. Please see below.

[1] Sanchez-Fernandez, A., Rumetshofer, E., Hochreiter, S., & Klambauer, G. (2022, March). Contrastive learning of image-and structure-based representations in drug discovery. In ICLR2022 Machine Learning for Drug Discovery.

---

### Decision · Program_Chairs · 2023-01-20

**Decision:**

Reject

**Justification For Why Not Higher Score:**

We did not reach a unanimous agreement on wether the positive or the negative arguments outweigh each other. Yet, as the authors have weakened their claim, the paper would require a significant re-write. Thus I now tend towards rejection on a very borderline paper.

**Justification For Why Not Lower Score:**

N/A

**Metareview: Summary, Strengths And Weaknesses:**

This paper proposes a cross-modal contrastive learning method that combines a cross-modal contrastive loss with several uni-modal losses to improve the analysis of cell images.

A Strength of the paper is the well-motivated and relevant application and the  comprehensive experimental results, that show that the paper  improves over the SOTA, esp on clinical outcome prediction tasks. The reviewers also find the paper is well-written and well-presented.

The main weaknesses are that the paper has limited novelty in terms of cross-modal alignment methods. The method also combines a larger number of loss functions, making it quite over-engineered.


**Summary Of Ac-Reviewer Meeting:**

On the call and the following discussion we went through the reviews and found that the main critique is the technical novelty of the method. This has not been sufficiently addressed during the rebuttal period. Rather, the authors change their claim in the paper improved the prior method by using additional loss terms.

On the other side, several reviewers found that the application is interesting to the reviewers and that it would make a good presentation. We looked at the call for papers, which indicates that applications of learning representation methods in biology are within scope.
The authors have provided additional experiments -- this effort is appreciated by all reviewers.

We did not reach a unanimous agreement on wether the positive or the negative arguments outweigh each other. Yet, as the authors have weakened their claim, the paper would require a significant re-write. Thus I now tend towards rejection on a very borderline paper.